# Efficient Conformal Prediction via Cascaded Inference with Expanded Admission

**Adam Fisch**[*]    **Tal Schuster**[*]    **Tommi Jaakkola**    **Regina Barzilay**
Computer Science and Artificial Intelligence Laboratory
Massachusetts Institute of Technology
`{fisch,tals,tommi,regina}@csail.mit.edu`

## Abstract

In this paper, we present a novel approach for conformal prediction (CP), in which we aim to identify a set of promising prediction candidates—in place of a single prediction. This set is guaranteed to contain a correct answer with high probability, and is well-suited for many open-ended classification tasks. In the standard CP paradigm, the predicted set can often be unusably large and also costly to obtain. This is particularly pervasive in settings where the correct answer is not unique, and the number of total possible answers is high. We first expand the CP correctness criterion to allow for additional, inferred "admissible" answers, which can substantially reduce the size of the predicted set while still providing valid performance guarantees. Second, we amortize costs by conformalizing prediction cascades, in which we aggressively prune implausible labels early on by using progressively stronger classifiers—again, while still providing valid performance guarantees. We demonstrate the empirical effectiveness of our approach for multiple applications in natural language processing and computational chemistry for drug discovery.[1]

## 1 Introduction

The ability to provide precise performance guarantees is critical to many classification tasks (Amodei et al., 2016; Jiang et al., 2012; 2018). Yet, achieving perfect accuracy with only single guesses is often out of reach due to noise, limited data, insufficient modeling capacity, or other pitfalls. Nevertheless, in many applications, it can be more feasible and ultimately as useful to hedge predictions by having the classifier return a *set* of plausible options—one of which is likely to be correct.

Consider the example of information retrieval (IR) for fact verification. Here the goal is to retrieve a snippet of text of some granularity (e.g., a sentence, paragraph, or article) that can be used to verify a given claim. Large resources, such as Wikipedia, can contain millions of candidate snippets—many of which may independently be able to serve as viable evidence. A good retriever should make precise snippet suggestions, quickly—but do so without excessively sacrificing sensitivity (i.e., recall).

Conformal prediction (CP) is a methodology for placing exactly that sort of bet on which candidates to retain (Vovk et al., 2005). Concretely, suppose we have been given $n$ examples, $(X_i, Y_i) \in \mathcal{X} \times \mathcal{Y}$, $i = 1, \ldots, n$, as training data, that have been drawn exchangeably from an underlying distribution $P$. For instance, in our IR setting, $X$ would be the claim in question, $Y$ a viable piece of evidence that supports or refutes it, and $\mathcal{Y}$ a large corpus (e.g., Wikipedia). Next, let $X_{n+1}$ be a new exchangeable test example (e.g., a new claim to verify) for which we would like to predict the paired $y \in \mathcal{Y}$. The aim of conformal prediction is to construct a set of candidates $\mathcal{C}_n(X_{n+1})$ likely to contain $Y_{n+1}$ (e.g., the relevant evidence) with *distribution-free marginal coverage* at a tolerance level $\epsilon \in (0, 1)$:

$$\mathbb{P}\left(Y_{n+1} \in \mathcal{C}_n(X_{n+1})\right) \geq 1 - \epsilon; \quad \text{for all distributions } P. \tag{1}$$

The marginal probability above is taken over all the $n + 1$ calibration and test points $\{(X_i, Y_i)\}_{i=1}^{n+1}$. A classifier is considered to be *valid* if the frequency of error, $Y_{n+1} \notin \mathcal{C}_n(X_{n+1})$, does not exceed $\epsilon$. In our IR setting, this would mean including the correct snippet at least $\epsilon$-fraction of the time. Not all valid classifiers, however, are particularly useful (e.g., a trivial classifier that merely returns all

---

[*]Equal contribution (author order decided randomly).
[1]Our code is available at `https://github.com/ajfisch/conformal-cascades`.

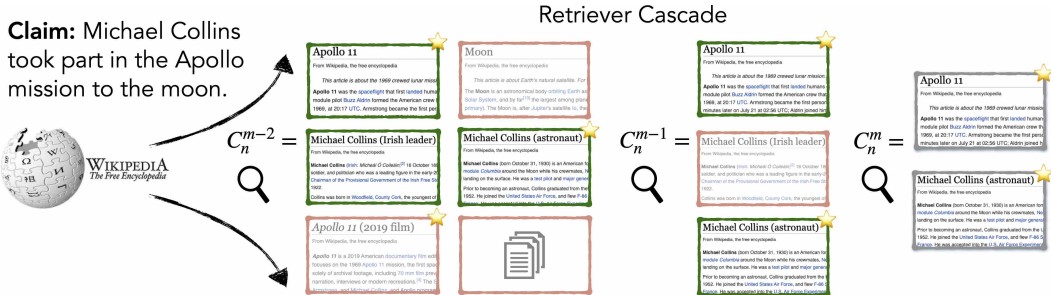

Figure 1: A demonstration of our conformalized cascade for $m$-step inference with set-valued outputs, here on an IR for claim verification task. The number of considered articles is reduced at every level—red frames are filtered, while green frames pass on. We only care about retrieving *at least one* of the admissible articles (starred) for resolving the claim.

possible outputs). A classifier is considered to have good *predictive efficiency* if $\mathbb{E}[|\mathcal{C}_n(X_{n+1})|]$ is small (i.e., $\ll |\mathcal{Y}|$). In our IR setting, this would mean not returning too many irrelevant articles—or in IR terms, maximizing precision while holding the level of recall at $\geq 1 - \epsilon$ (assuming $Y$ is a single answer). In practice, in domains where the number of outputs to choose from is large and the "correct" one is not necessarily unique, classifiers derived using conformal prediction can suffer dramatically from both poor predictive and computational efficiency (Burnaev and Vovk, 2014; Vovk et al., 2016; 2020). Unfortunately, these two conditions tend to be compounding: large label spaces $\mathcal{Y}$ both (1) often place strict constraints on the set of tractable model classes available for consideration, and (2) frequently contain multiple clusters of labels that are difficult to discriminate between, especially for a low-capacity classifier.

In this paper, we present two effective methods for improving the efficiency of conformal prediction for classification tasks with large output spaces $\mathcal{Y}$, in which several $y \in \mathcal{Y}$ might be admissible—i.e., acceptable for the purposes of our given task. First, in Section 4 we describe a generalization of Eq. 1 to an expanded admission criteria, where $\mathcal{C}_n(X_{n+1})$ is considered valid if it contains at least one admissible $y$ with high probability. For example, in our IR setting, given the claim "*Michael Collins took part in the Apollo mission to the moon*," any of the articles "*Apollo 11*," "*Michael Collins (astronaut)*," or "*Apollo 11 (2019 film)*" have enough information to independently support it (see Figure 1)—and are therefore all admissible. When $Y_{n+1}$ is not unique, forcing the classifier to hedge for the worst case, in which a specific realization of $Y_{n+1}$ must be contained in $\mathcal{C}_n(X_{n+1})$, is too strict and can lead to conservative predictions. We theoretically and empirically show that optimizing for an expanded admission criteria yields classifiers with significantly better predictive efficiency.

Second, in Section 5 we present a technique for conformalizing *prediction cascades* to progressively filter the number of candidates with a sequence of increasingly complex classifiers. This allows us to balance predictive efficiency with computational efficiency during inference. Importantly, we also theoretically show that, in contrast to other similarly motivated pipelines, our method filters the output space in a manner that still guarantees marginal coverage. Figure 1 illustrates our combined approach. We demonstrate that, together, these two approaches serve as complementary pieces of the puzzle towards making CP more efficient. We empirically validate our approach on information retrieval for fact verification, open-domain question answering, and in-silico screening for drug discovery.

**Contributions.** In summary, our main results are as follows:

- A theoretical extension of validity (Eq. 1) to allow for inferred *admissible* answers.

- A principled framework for conformalizing computationally efficient prediction cascades.

- Consistent empirical gains on three diverse tasks demonstrating up to $4.6\times$ better predictive efficiency AUC (measured across all $\epsilon$) when calibrating for expanded admission, with computation pruning factors of up to $1/m$, where $m$ is the number of models, when using prediction cascades.

## 2   RELATED WORK

**Confident prediction.** Methods for obtaining precise uncertainty estimates have received intense interest in recent years. A significant body of work is concerned with *calibrating* model confidence—

measured as $p_\theta(\hat{y}_{n+1}|x_{n+1})$—such that the true accuracy, $y_{n+1} = \hat{y}_{n+1}$, is indeed equal to the estimated probability (Niculescu-Mizil and Caruana, 2005; Gal and Ghahramani, 2016; Lakshminarayanan et al., 2017; Lee et al., 2018). In theory, these estimates could be leveraged to create confident prediction sets $\mathcal{C}_n(X_{n+1})$. Unlike CP, however, these methods are not guaranteed to be accurate, and often still suffer from miscalibration in practice—especially for modern neural networks (Guo et al., 2017; Ashukha et al., 2020; Hirschfeld et al., 2020). *Selective classification* (El-Yaniv and Wiener, 2010; Geifman and El-Yaniv, 2017), where models have the option to abstain from answering when not confident, is similar in motivation to Eq. 1. In fact, it can be considered as a special case in which the classifier chooses to abstain unless $|\mathcal{C}_n(X_{n+1})| = 1$.

**Conformal prediction.** As validity is already guaranteed by design in conformal prediction, most efforts in CP focus on improving various aspects of efficiency. Mondrian CP (Vovk et al., 2005) accounts for the fact that some classes are harder to model than others, and leverages class-conditional statistics. Similiarly, several recent studies have built towards conditional—as opposed to marginal—coverage through various adaptive approaches, such as conformalizing quantile functions or working with conditional distributions that vary with $x$ (see Cauchois et al., 2020; Chernozhukov et al., 2019; Kivaranovic et al., 2020; Romano et al., 2019; 2020, *inter alia*). Cauchois et al. (2020) also directly model dependencies among $y$ variables for use in multi-label prediction. Our method for *expanded admission*, on the other hand, aggregates statistics for equivalent *single* labels by example and across classes. Though we only provide marginal guarantees, the ideas expressed in those related works are complementary, and can be applied here as well. Inductive CP (Papadopoulos, 2008) is also complementary extension that dramatically reduces the cost of computing $\mathcal{C}_n(X_{n+1})$ in the general case; we make use of it here. Most similar to our work, trimmed (Chen et al., 2016) and discretized (Chen et al., 2018) CP trade *predictive* efficiency for *computational* efficiency in regression tasks, where the label space is infinite. A key distinction of our method is that we do not force the same trade-off: in fact, we empirically show that our conformalized cascades can at times result in *better* predictive efficiency alongside a pruned label space.

**Prediction cascades.** The idea of balancing cost with accuracy by using multi-step inference has been explored extensively for many applications (Charniak et al., 2006; Deng and Rush, 2020; Fleuret and Geman, 2001; Jurafsky and Martin, 2000; Li et al., 2015; Rush and Petrov, 2012). Some of these methods use fixed rules with no performance guarantees, such as greedy pipelines where the top $k$ predictions are passed on to the next level (Chen et al., 2017; Ferrucci et al., 2010). Closer to our work, Weiss and Taskar (2010) optimize their cascades for overall pruning efficiency, and not for top-1 prediction. While they also analyze error bounds for filtering, their guarantees are specific to linear classifiers with bounded $L_2$ norm, whereas our conformalized approach only assumes data exchangeability. Furthermore, they assume a target filtering loss before training—our tolerance level $\epsilon$ is defined at test time, which allows for much greater flexibility.

## 3 BACKGROUND

We begin with a brief review of conformal prediction (see Shafer and Vovk, 2008). Here, and in the rest of the paper, upper-case letters ($X$) denote random variables; lower-case letters ($x$) denote scalars, and script letters ($\mathcal{X}$) denote sets, unless otherwise specified. Proofs are deferred to the appendix.

At the core of conformal prediction is a simple statistical hypothesis test: for each candidate $y \in \mathcal{Y}$ we must either accept or reject the null hypothesis that $(X_{n+1} = x_{n+1}, Y_{n+1} = y)$ is a correct pairing. Formally, we rely on a *nonconformity measure* $\mathcal{S}\big((x_{n+1}, y), \mathcal{D}\big)$ to serve as the test statistic, where a higher value of $\mathcal{S}$ reflects that $(x_{n+1}, y)$ is less "conforming" to the distribution specified by dataset $\mathcal{D}$. For instance, $\mathcal{S}$ could be computed as $-\log p_\theta(y|x)$, where $\theta$ is a model fit to $\mathcal{D}$.

**Definition 3.1** (Nonconformity measure). Let $\mathcal{Z} := \mathcal{X} \times \mathcal{Y}$ be the space of examples $(X, Y)$, and let $\mathcal{Z}^{(*)} := \bigcup_{d \geq 1}(\mathcal{X} \times \mathcal{Y})^d$ be the space of datasets of examples $\mathcal{D}$, of any size $d \geq 1$. A *nonconformity measure* $\mathcal{S}$ is then a measurable mapping $\mathcal{S} \colon \mathcal{Z} \times \mathcal{Z}^{(*)} \to \mathbb{R}$, that assigns a real-valued score to any example $(X, Y)$, indicating how different[2] it is from a reference dataset $\mathcal{D}$. Furthermore, in order to retain exchangeability, $\mathcal{S}$ is symmetric with respect to permutations of its input data.

To be specific, exact or *full* CP takes $\mathcal{D}$ to be all of the examples seen so far, including the candidate $(x_{n+1}, y)$. Thus, the nonconformity measure $\mathcal{S}$ has to be re-trained each time. An alternative—which

---

[2]The definition of "different" here is intentionally vague, as any metric will technically work.

we use in this paper *w.l.o.g.*—is the inductive or *split* CP variant (Papadopoulos, 2008) which assumes that $\mathcal{D}$ is a proper training set, independent of any of the subsequent $n + 1$ exchangeable examples used for CP. Dropping $\mathcal{D}$ for ease of notation, we denote the score for example $(X, Y)$ as the random variable $\mathcal{S}(X, Y)$. The degree of nonconformity can then be quantified using a p-value.

**Lemma 3.2** (Smoothed p-value). *Assume that the random variables $V_1, \ldots, V_{n+1}$ are exchangeable. We define the smoothed empirical p-value* $\mathtt{pvalue}(V_{n+1}, V_{1:n})$ *as*

$$\mathtt{pvalue}(V_{n+1}, V_{1:n}) := \frac{|\{i \in [1, n] \colon V_i > V_{n+1}\}| + \tau \cdot |\{i \in [1, n] \colon V_i = V_{n+1}\}| + 1}{n + 1}, \quad (2)$$

*where $\tau \sim U(0, 1)$. Then, for any $\epsilon \in (0, 1)$, we have $\mathbb{P}(\mathtt{pvalue}(V_{n+1}, V_{1:n}) \leq \epsilon) \leq \epsilon$.*

To construct the final conformal *prediction*, the classifier uses the p-values to include all $y$ for which the null hypothesis—i.e., that the candidate pair $(x_{n+1}, y)$ is *conformal*—is not rejected.

**Theorem 3.3** (CP; Vovk et al. (2005), see also Lei et al. (2018)). *Assume that the random variables $(X_i, Y_i) \in \mathcal{X} \times \mathcal{Y}$, $i = 1, \ldots, n + 1$ are exchangeable. For any nonconformity measure $\mathcal{S}$, and $\epsilon \in (0, 1)$, define the conformal label set (based on the first n samples) at $x_{n+1} \in \mathcal{X}$ as*

$$\mathcal{C}_n(x_{n+1}) := \Big\{ y \in \mathcal{Y} \colon \mathtt{pvalue}\big(\mathcal{S}(x_{n+1}, y), \mathcal{S}(x_{1:n}, y_{1:n})\big) > \epsilon \Big\}. \quad (3)$$

*Then $\mathcal{C}_n(X_{n+1})$ satisfies Eq. 1, where $\mathbb{P}\left(Y_{n+1} \in \mathcal{C}_n(X_{n+1})\right) \geq 1 - \epsilon$.*

## 4 CONFORMAL PREDICTION WITH EXPANDED ADMISSION

We now introduce our strategy for improving the *predictive efficiency* of CP classifiers. What might it mean for an alternative label $y$ to be "good enough?" Among other factors, this depends on the task, the label space $\mathcal{Y}$, and even the input $x$. For example, in IR, two different texts might independently provide sufficient information for claim $x$ to be resolved. Formally, we pose the underlying setting as a set-valued function $f : \mathcal{X} \to 2^{\mathcal{Y}}$, where the ground truth is defined as the expanded set of all *admissible* answers $f(X)$ for input $X$ (e.g., given our notions of semantic equivalence or error tolerance). Unlike ranking or multi-label classification, however, our evaluation only demands retrieving a single element of this ground truth set (and without any preference as to which element).

It is often the case that this underlying function $f$ remains unknown—after all, exhaustively annotating *all* possible admissible answers can quickly become intractable for many problems. Rather, we assume that what we observe in our dataset are samples, $(X_i, Y_i)$, from the underlying ground truth sets $f(X_i)$ via some additional observation process (e.g., influenced by which annotator wrote the answer to the question). In this view, the distribution $P$ governing each pair $(X_i, Y_i)$ is an induced distribution from this set-valued function, together with the unknown observation process. We can then use the provided dataset reference $Y_i$ to seed a label-expansion operation, in an attempt to approximate $f(X_i)$. More concretely, for some choice of admission function $g \colon (\mathcal{X} \times \mathcal{Y}) \times \mathcal{Y} \to \{0, 1\}$, we construct a set of inferred admissible labels $\mathcal{A}_g$ given the seed reference $(X = x, Y = y)$, i.e.,

$$\mathcal{A}_g(x, y) := \Big\{ \bar{y} \in \mathcal{Y} \colon g(x, y, \bar{y}) = 1 \Big\}. \quad (4)$$

**Assumption 4.1.** For any reference label $Y \in f(X)$, we have that $\mathcal{A}_g(X, Y)$ contains $Y$ and is a subset of the full ground truth. That is, $\mathcal{A}_g(X, Y)$ obeys $Y \in \mathcal{A}_g(X, Y) \subseteq f(X)$.

In this work we assume that $g$ is given to us (not learned), and is a deterministic function that has no inherent error in its outputs. For many tasks, this is a quite natural assumption, as it is often feasible for the user to define a set of rules that qualify label admission—e.g., syntactic normalization rules in NLP, or expanding some small $\delta$-neighborhood of the original $y$ given some metric. In Appendix B we discuss a relaxation of this assumption in which $\mathcal{A}_g(X, Y)$ might have some error.

Given $g$, $x$, and $y$, any prediction that is a member of the derived set $\mathcal{A}_g$ is then considered to be *admissible*, i.e., a success. For example, in IR for claim verification, let $h$ be the verification model (or a human) that, given the claim $x$ and evidence $y$, outputs a score for the final verdict (that the claim is true or false). The admission might then be defined as $g(x, y, \bar{y}) := \mathbf{1}\{|h(y, x) - h(\bar{y}, x)| \leq \delta\}$, where $\delta$ is a small slack parameter. This then leads us to a helpful definition of expanded admission:

**Definition 4.2** (Expanded admission). Given any label admission function $g$ and data points $\{(X_i, Y_i)\}_{i=1}^n$ drawn exchangeably from an underlying distribution $P$, a conformal predictor pro-

---

**Algorithm 1** Cascaded inductive conformal prediction with distribution-free marginal coverage.

---

**Definitions:** $(\mathcal{S}_1, \ldots, \mathcal{S}_m)$ is a sequence of nonconformity measures. $\mathcal{M}$ is a monotonic correction controlling for family-wise error. $x_{n+1} \in \mathcal{X}$ is the given test point. $x_{1:n} \in \mathcal{X}^n$ and $y_{1:n} \in \mathcal{Y}^n$ are the previously observed calibration examples and their labels, respectively. $\mathcal{Y}$ is the label space. $\epsilon$ is the tolerance level.

1: **function** PREDICT$(x_{n+1}, (x_{1:n}, y_{1:n}), \epsilon)$
2:      $\mathcal{C}_n^0 \leftarrow \mathcal{Y}$          ▷ Initialize with the full label set.
3:      $p_1^{(y)} = p_2^{(y)} = \ldots = p_m^{(y)} \leftarrow 1, \forall y \in \mathcal{Y}$      ▷ Conservatively set unknown p-values.
4:      **for** $j = 1$ to $m$ **do**
5:          $\mathcal{C}_n^j \leftarrow \{\}$      ▷ Initialize the current output.
6:          **for** $y \in \mathcal{C}_n^{j-1}$ **do**      ▷ Iterate through the previous label set.
7:              $p_j^{(y)} \leftarrow \texttt{pvalue}\left(\mathcal{S}_j(x_{n+1}, y), \ \mathcal{S}_j(x_{1:n}, y_{1:n})\right)$    ▷ Update the $j$-th p-value for $(x_{n+1}, y)$.
8:              $\tilde{p}_j^{(y)} \leftarrow \mathcal{M}(p_1^{(y)}, \ldots, p_m^{(y)})$      ▷ Correct the current p-values for MHT.
9:              **if** $(\tilde{p}_j^{(y)} > \epsilon)$ **then**      ▷ Keep $y$ iff the corrected p-value supports it.
10:                  $\mathcal{C}_n^j \leftarrow \mathcal{C}_n^j \cup \{y\}$
11:      **return** $\mathcal{C}_n^m$      ▷ Return the final output of the cascade.

---

ducing admissible predictions $\mathcal{C}_n(X_{n+1})$ (based on the first $n$ points) for a new exchangeable test example $X_{n+1}$ is considered to be valid under *expanded admission* if for any $\epsilon \in (0,1)$, $\mathcal{C}_n$ satisfies

$$\mathbb{P}\Big(|\mathcal{A}_g(X_{n+1}, Y_{n+1}) \cap \mathcal{C}_n(X_{n+1})| \geq 1\Big) \geq 1 - \epsilon; \quad \textit{for all distributions } P. \tag{5}$$

Recall that by *predictive efficiency* we mean that we desire $|\mathcal{C}_n(X_{n+1})|$ to be small. A CP that simply returns all possible labels is trivially valid, but not useful. By Definition 4.2, we only need to identify *at least one* $\bar{Y}_{n+1}$ that is deemed admissible according to $g$. As such, we propose a modification that allows the classifier to be more discriminative—and produce smaller $\mathcal{C}_n(X_{n+1})$—when testing the null hypothesis that $y$ is not just conforming, but that it is the *most conforming* admissible $y$ for $x$. For each data point $(X_i = x_i, Y_i = y_i)$, we then define the minimal nonconformity score as

$$\mathcal{S}_g^{\min}(x_i, y_i) := \min\Big\{\mathcal{S}(x_i, \bar{y}) \colon \bar{y} \in \mathcal{A}_g(x_i, y_i)\Big\}, \tag{6}$$

and use these random variables to compute the p-values for our conformal predictor.

**Theorem 4.3** (CP with expanded admission). *Assume that $(X_i, Y_i) \in \mathcal{X} \times \mathcal{Y}$, $i = 1, \ldots, n+1$ are exchangeable. For any non-conformity measure $\mathcal{S}$, label admission function $g$, and $\epsilon \in (0,1)$, define the conformal set (based on the first $n$ samples) at $x_{n+1} \in \mathcal{X}$ as*

$$\mathcal{C}_n^{\min}(x_{n+1}) := \Big\{ y \in \mathcal{Y} \colon \texttt{pvalue}\left(\mathcal{S}(x_{n+1}, y), \mathcal{S}_g^{\min}(x_{1:n}, y_{1:n})\right) > \epsilon \Big\}. \tag{7}$$

*Then $\mathcal{C}_n^{\min}(X_{n+1})$ satisfies Eq. 5. Furthermore, we have $\mathbb{E}\left[|\mathcal{C}_n^{\min}(X_{n+1})|\right] \leq \mathbb{E}\left[|\mathcal{C}_n(X_{n+1})|\right]$.*

We present a proof in Appendix A.3. In Section 7 we demonstrate empirically that this simple modification can yield large improvements in efficiency, while still maintaining the desired coverage.

## 5   CONFORMAL PREDICTION CASCADES

We now introduce our strategy for improving the *computational efficiency* of conformal prediction. As is apparent from Eq. 3, the cost of conformal prediction is linear in $|\mathcal{Y}|$. In practice, this can limit the tractable choices available for the nonconformity measure $\mathcal{S}$, particularly in domains where $|\mathcal{Y}|$ is large. Furthermore, predictive and computational efficiency are coupled, as being forced to use weaker $\mathcal{S}$ reduces the statistical power of the CP. Our approach balances the two by leveraging prediction cascades (Sapp et al., 2010; Weiss and Taskar, 2010, *inter alia*), where $m$ models of increasing power are applied sequentially. At each stage, the number of considered outputs is iteratively pruned. Critically, we *conformalize* the cascade, which preserves marginal coverage. For clarity of presentation, in this section we return to the standard setting without expanded admission (i.e., where $\mathcal{C}_n(X_{n+1})$ satisfies Eq. 1), but emphasize that the method applies to either case.

When constructing $\mathcal{C}_n(X_{n+1})$ via Eq. 3, a nonconformity score and corresponding p-value is computed for every candidate $y \in \mathcal{Y}$. Different $y$, however, might be much easier to reject than others, and can be filtered using simple metrics. For example, in IR, wholly non-topical sentences (of which there are many) can be discarded using fast key-word matching algorithms such as TFIDF or BM25. On the other hand, more ambiguous sentences—perhaps those on the same topic but with insufficient information—might require a more sophisticated scoring mechanism, such as a neural network.

Assume that we are given a sequence of progressively more discriminative, yet also more computationally expensive, nonconformity measures $(\mathcal{S}_1, \ldots, \mathcal{S}_m)$. When applied in order, we only consider $y \in \mathcal{C}_n^i(X_{n+1})$ as candidates for inclusion in $\mathcal{C}_n^{i+1}(X_{n+1})$. Thus, $\mathcal{C}_n^m(X_{n+1}) \subseteq \mathcal{C}_n^{m-1}(X_{n+1}) \subseteq \ldots \subseteq \mathcal{C}_n^1(X_{n+1}) \subseteq \mathcal{Y}$. In this way, the amortized cost of evaluating $m$ measures over *parts* of $\mathcal{Y}$ can be lower than the cost of running one expensive measure over *all* of it. For example, in IR, we can use BM25 ($\mathcal{S}_1$) to prune the label space passed to a neural model ($\mathcal{S}_2$). Furthermore, combining multiple nonconformity measures together can also lead to better predictive efficiency when using complementary measures—similar to ensembling (Toccaceli and Gammerman, 2017).

Naïvely applying multiple tests to the same data, however, leads to the *multiple hypothesis testing* (MHT) problem. This results in an increased *family-wise error rate* (i.e., false positives), making the CP invalid. Many corrective procedures exist in the literature (e.g., see Liu (1996)). Formally, given $m$ p-values $(P_1, \ldots, P_m)$ for a pair $(X, Y)$, we denote as $\mathcal{M}$ some such correction satisfying

$$\tilde{P} = \mathcal{M}(P_1, \ldots, P_m) \quad \text{s.t.} \quad \mathbb{P}\left(\tilde{P} \leq \epsilon \mid Y \text{ is correct}\right) \leq \epsilon. \tag{8}$$

Furthermore, we require $\mathcal{M}$ to be element-wise monotonic[3], i.e. (where $\preccurlyeq$ operates element-wise):

$$(P_1, \ldots, P_m) \preccurlyeq \left(\hat{P}_1, \ldots, \hat{P}_m\right) \implies \mathcal{M}(P_1, \ldots, P_m) \leq \mathcal{M}\left(\hat{P}_1, \ldots, \hat{P}_m\right). \tag{9}$$

We consider several options for $\mathcal{M}$, namely the Bonferroni and Simes corrections (see Appendix E). In order to exit the test early at cascade $j$ before all the p-values (i.e., for measures $k > j$) are known, we compute an upper bound for the corrected p-value by conservatively assuming that $P_k = 1$, $\forall k > j$. The full procedure is demonstrated in Algorithm 1, and formalized in Theorem 5.1.

**Theorem 5.1** (Cascaded CP). *Assume that $(X_i, Y_i) \in \mathcal{X} \times \mathcal{Y}$, $i = 1, \ldots, n+1$ are exchangeable. For any sequence of nonconformity measures $(\mathcal{S}_1, \ldots, \mathcal{S}_m)$ yielding p-values $(P_1, \ldots, P_m)$, and $\epsilon \in (0, 1)$, define the conformal set for step $j$ (based on the first $n$ samples) at $x_{n+1} \in \mathcal{X}$ as*

$$\mathcal{C}_n^j(x_{n+1}) := \left\{y \in \mathcal{Y} \colon \tilde{P}_j^{(y)} > \epsilon\right\}, \tag{10}$$

*where $\tilde{P}_j^{(y)}$ is the conservative p-value for candidate $y$ at step $j$, $\mathcal{M}(P_1^{(y)}, \ldots, P_j^{(y)}, 1, \ldots, 1)$, with $P_{k>j}^{(y)} := 1$. Then $\forall j \in [1, m]$, $\mathcal{C}_n^j(X_{n+1})$ satisfies Eq. 1, and $\mathcal{C}_n^m(X_{n+1}) \subseteq \mathcal{C}_n^j(X_{n+1})$.*

We present a proof in Appendix A.4. Theorem 5.1 also easily extends to the setting of Eq. 5. An important result is that early pruning will *not* affect the validity of the final result, $\mathcal{C}_n^m$.

## 6 EXPERIMENTAL SETUP

We empirically evaluate our method on three different tasks with standard, publicly available datasets. In this section, we briefly give a high-level outline of each task and our conformalized approach to it. We also describe our evaluation methodology. We defer the technical details for each task, such as data preprocessing, training, and nonconformity measure formulations, to Appendix C.

### 6.1 TASKS

**Open-domain question answering (QA).** Open-domain question answering focuses on using a large-scale corpus $\mathcal{D}$ to answer arbitrary questions via search combined with reading comprehension. We use the open-domain setting of the Natural Questions dataset (Kwiatkowski et al., 2019). Following Chen et al. (2017), we first retrieve relevant passages from Wikipedia using a document retriever, and then select an answer span from the considered passages using a document reader. We use a Dense Passage Retriever model (Karpukhin et al., 2020) for the retriever, and a BERT model (Devlin

---

[3]Note that we are unaware of any common $\mathcal{M}$ beyond contrived examples satisfying (8) but not (9).

et al., 2019) for the reader. The BERT model yields several score variants—we use multiple in our cascade (see Table C.1). Any span from any retrieved passage that matches any of the annotated answer strings when lower-case and stripped of articles and punctuation is considered to be correct.

**Information retrieval for fact verification (IR).** As introduced in §1, the goal of IR for fact verification is to retrieve a sentence that can be used to support or refute a given claim. We use the FEVER dataset (Thorne et al., 2018), in which evidence is sourced from a set of ∼40K sentences collected from Wikipedia. A sentence that provides enough evidence for the correct verdict (true/false) is considered to be acceptable (multiple are labeled in the dataset). Our cascade consists of (1) a fast, non-neural BM25 similarity score between a given claim and sentence, and (2) the score of an ALBERT model (Lan et al., 2020) trained to directly predict if a given claim and sentence are related.

**In-silico screening for drug discovery (DR).** In-silico screening of chemical compounds is a common task in drug discovery/repurposing, where the goal is to identify possibly effective drugs to manufacture and test (Stokes et al., 2020). Using the ChEMBL database (Mayr et al., 2018), we consider the task of screening molecules for combinatorial constraint satisfaction, where given a specified constraint such as "*has property A but not property B*", we want to identify at least one molecule from a given set of candidates that has the desired attributes. Our cascade consists of (1) the score of a fast, non-neural Random Forest (RF) applied to binary Morgan fingerprints (Rogers and Hahn, 2010), and (2) the score of a directed Message Passing NN ensemble (Yang et al., 2019).

### 6.2 EVALUATION METRICS

For each task, we use a proper training, validation, and test set. We use the training set to learn all nonconformity measures $\mathcal{S}$. We perform model selection specifically for CP on the validation set, and report final numbers on the test set. For all CP methods, we report the marginalized results over 20 random trials, where in each trial we partition the data into 80% calibration ($x_{1:n}$) and 20% prediction points ($x_{n+1}$). In order to compare the aggregate performance of different CPs across all tolerance levels, we plot each metric as a function of $\epsilon$, and compute the area under the curve (AUC). In all plots, shaded regions show the 16-84th percentiles across trials. We use the following metrics:

**Predictive accuracy.** We measure the accuracy rate as the rate at which at least one admissible prediction is in $\mathcal{C}_n$, i.e., $|\mathcal{A}_g(X_{n+1}, Y_{n+1}) \cap \mathcal{C}_n(X_{n+1})| \geq 1$ (see Eq. 5). To be valid—the key criteria in this work—a classifier should have an accuracy rate $\geq 1 - \epsilon$, and AUC $\geq 0.5$. Note that *more* is not necessarily *better*: higher success rates than required can lead to poor efficiency (i.e., the size of $\mathcal{C}_n$ can afford to decrease at the expense of accuracy).

**Predictive efficiency (↓).** We measure predictive efficiency as the size of the prediction set out of all candidates: $|\mathcal{C}_n| \cdot |\mathcal{Y}|^{-1}$. The goal is to make the predictions more precise while still maintaining validity. *Lower* predictive efficiency is better (↓), as it means that the size of $\mathcal{C}_n$ is relatively *smaller*.

**Amortized computation cost (↓).** We measure the amortized computation cost as the ratio of `pvalue` computations required to make a cascaded prediction with early pruning, compared to when using a simple combination of CPs (no pruning, same number of measures). In this work we do not measure wall-clock times as these are hardware-specific, and depend heavily on optimized implementations. *Lower* amortized cost is better, as it means that the relative number of p-value computations required to construct $\mathcal{C}_n^m$ (for a given $m$) is *smaller*.

## 7 EXPERIMENTAL RESULTS

In the following, we address several key research questions relating to our two combined conformal prediction advancements, and their impact on overall performance. In all of the following experiments, we report results on the test set, using cascade configurations selected based on the validation set performance. The QA and IR cascades use the Simes correction for MHT, while the DR cascades uses the Bonferroni correction. Additional results, details, and analysis are included in Appendix D.

**Predictive efficiency of expanded admission.** We begin by testing how deliberately calibrating for expanded admission affects predictive efficiency. That is, we use minimal admissible nonconformity scores $\{\mathcal{S}_g^{\min}(X_i, Y_i)\}_{i=1}^n$ for calibration (Eq. 6) to create $\mathcal{C}_n(X_{n+1})$ using Eq. 7. Figure 2 shows the predictive efficiency of our *min*-calibrated CPs with expanded admission across all $\epsilon$, while Table 1 shows results for select values of $\epsilon$. We compare both cascaded and non-cascaded CPs, as Theorem 4.3 applies to both settings. The non-cascaded CP uses the last nonconformity measure

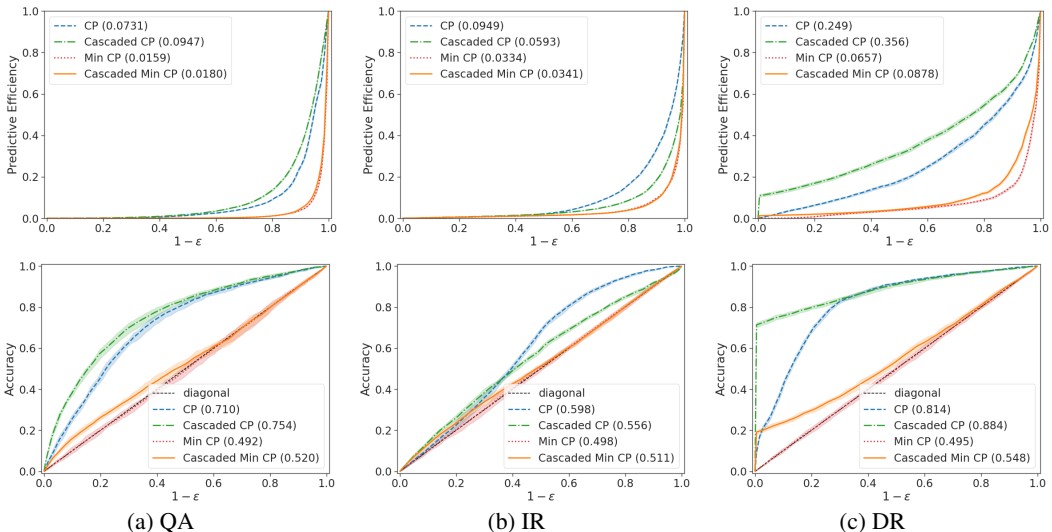

(a) QA   (b) IR   (c) DR

Figure 2: *Does calibrating using minimal admissible nonconformity scores (§4) improve predictive efficiency?* We show predictive efficiency and success rates as a function of $\epsilon$. The efficiency of the *min*-calibrated CPs (both cascaded and non-cascaded) is significantly better (i.e., lower) than standard CP across all tasks, especially at critical values of $\epsilon$. The *min*-calibrated CP's accuracy hugs the diagonal, allowing it to remain valid, but be far less conservative (resulting in better efficiency).

| Task | Target Acc. $(1-\epsilon)$ | Baseline CP | | *min*-CP | | Cascaded *min*-CP | | |
|------|------|------|------|------|------|------|------|------|
| | | Acc. | $|\mathcal{C}_n|$ | Acc. | $|\mathcal{C}_n^{\min}|$ | Acc. | $|\mathcal{C}_n^{\min}|$ | Amortized cost |
| **QA** | 0.90 | 0.98 | 1245.7 | 0.90 | 198.0 | 0.90 | 235.1 | 0.59 |
| | 0.80 | 0.94 | 453.5 | 0.80 | 58.7 | 0.80 | 57.7 | 0.50 |
| | 0.70 | 0.91 | 227.8 | 0.70 | 22.1 | 0.70 | 20.6 | 0.45 |
| | 0.60 | 0.87 | 127.8 | 0.60 | 10.6 | 0.61 | 9.5 | 0.42 |
| **IR** | 0.99 | 1.00 | 33.6 | 0.99 | 17.8 | 0.99 | 18.0 | 0.92 |
| | 0.95 | 1.00 | 20.9 | 0.95 | 7.4 | 0.95 | 7.4 | 0.79 |
| | 0.90 | 0.98 | 13.8 | 0.90 | 4.0 | 0.90 | 3.8 | 0.73 |
| | 0.80 | 0.95 | 6.7 | 0.80 | 1.7 | 0.80 | 1.6 | 0.63 |
| **DR** | 0.90 | 0.99 | 429.8 | 0.90 | 84.1 | 0.91 | 147.8 | 0.84 |
| | 0.80 | 0.97 | 305.8 | 0.80 | 42.7 | 0.81 | 62.1 | 0.79 |
| | 0.70 | 0.96 | 216.6 | 0.70 | 28.1 | 0.71 | 37.3 | 0.75 |
| | 0.60 | 0.94 | 145.6 | 0.60 | 19.3 | 0.63 | 24.6 | 0.72 |

Table 1: CP results for specific tolerance levels $\epsilon$. Each line shows the empirical accuracy (Acc.) and the (raw) size of the prediction set for our two methods as compared to regular CP, per target accuracy rate $(1-\epsilon)$. The amortized computation cost of the cascade, a consequence of pruning, is also given.

of the cascaded CP. Across all tasks, the efficiency of the *min*-calibrated CP is significantly better than the baseline CP method, and results in tighter prediction sets—giving up to $4.6\times$ smaller AUC. Naturally, this effect depends on the qualities of the admission function $g$ and resulting admissible label sets $\mathcal{A}_g$. For example, the most dramatic gains are seen on QA. This task has a relatively large variance among admissible label nonconformity scores (i.e., some admissible answer spans are much easier to identify than others), and thus calibrating for the most conforming spans has a large impact.

**Validity of minimal nonconformity calibration.** As per Theorem 4.3, we observe that our *min*-calibrated CP still creates valid predictors, with an accuracy rate close to $1-\epsilon$ on average. We see that the *min*-calibration allows the predictor to reject more wrong predictions (lower predictive efficiency, which is better) while, with high enough probability, still accept at least one that is correct. The standard CP methods, however, are more conservative—and result in prediction sets and accuracy rates larger than necessary. This effect is most pronounced at smaller $\epsilon$ (e.g., $< 0.2$).

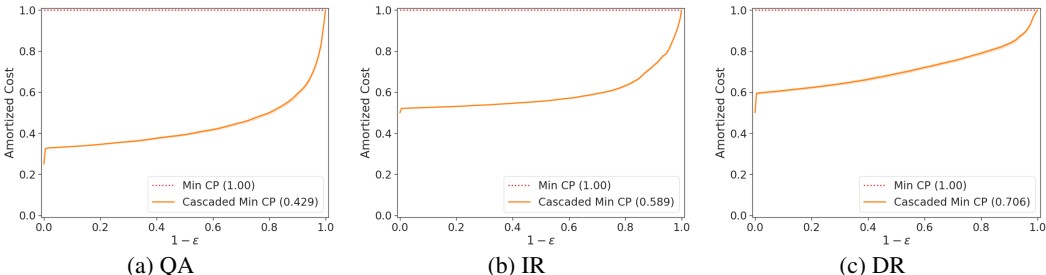

Figure 3: *How effective is cascaded conformal prediction (§5) at pruning the candidate space?* Incorporating early rejection via the CP cascade reduces the fraction of required p-value computations. Larger tolerance levels ($\epsilon$) allow for more aggressive pruning rates.

**Computational efficiency of conformalized cascades.** Figure 3 shows the amortized cost, in terms of percentage of p-value computations skipped, achieved using our cascaded CP algorithm. Our method reduces the number of p-value computations by up to a factor of $1/m$ for an $m$-layer cascade. The effect of conformalization is clear: the more strict an $\epsilon$ we demand, the fewer labels we can prune, and vice versa. To simplify comparisons, our metric gives equal weight to all p-value computations. In practice, however, the benefits of early pruning will generally grow by layer, as the later p-values are assumed to be increasingly expensive to compute. Again, exactly quantifying this trade-off in terms of absolute cost is model- and implementation-specific; we leave it for future work.

**Conservativeness of conformalized cascades.** A limitation of the cascade approach is the conservative nature of the MHT-corrected p-value (i.e., the Bonferroni effect), which can reduce the statistical power of the CP as the number of cascade layers grows. This effect is especially present if the cascaded measures are highly dependent. In general, however, in both Figure 2 and Table 1, we see that the benefits of combining complementary cascaded models largely make up for this drop in statistical power, as our cascaded *min*-calibrated CPs nearly matches the predictive efficiency of our non-cascaded models. Importantly, this is achieved while still improving computational efficiency.

**Relation to heuristic methods.** For completeness, in Appendix D.2 we also compare CP to common heuristic methods for producing set-valued predictions—namely, taking the top-$k$ predictions and taking all predictions for which the model's score exceeds a threshold $\tau$. We show that while CP is more general, it can be (practically) reduced to each of these methods with the appropriate choice of nonconformity measure. In some cases, the flexibility of CP even allows for better predictive efficiency, even while those heuristics do not amortize cost or guarantee coverage in finite samples.

## 8 CONCLUSION

Conformal prediction can afford remarkable theoretical performance guarantees to important applications for which high accuracy and precise confidence estimates are key. Naively applying CP, however, can be inefficient in practice. This is especially true in realistic domains in which the correct answers are not clearly delineated, and in which the computational cost of discriminating between options starts to become a limiting factor. In this paper, we proposed two novel methods that provide two more pieces of the puzzle. Our results show that (1) calibration using expanded admission consistently improves empirical predictive efficiency, and (2) conformal prediction cascades yield better computational efficiency—and thereby enable the use of more powerful classifiers.

## ACKNOWLEDGEMENTS

We specially thank Ben Fisch for his helpful comments and discussion on this work, in addition to Kyle Swanson for help in running the `chemprop` models. We also thank the MIT NLP group for valuable feedback. AF is supported in part by the National Science Foundation Graduate Research Fellowship under Grant #1122374. Any opinion, findings, conclusions, or recommendations expressed in this material are those of the authors and do not necessarily reflect the views of the the NSF or the U.S. Government. TS is supported in part by DSO grant DSOCL18002. This work is also supported in part by MLPDS and the DARPA AMD project.

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

# Appendix

## A PROOFS

### A.1 PROOF OF LEMMA 3.2

*Proof.* This is a well-known result; we prove it here for completeness (see also Tibshirani et al. (2019) for a similar proof). It is straightforward to show that for $P = \texttt{pvalue}(V_{n+1}, V_{1:n})$,

$$P \leq \epsilon \Longleftrightarrow V_{n+1} \text{ is ranked among the } \lfloor \epsilon \cdot (n+1) \rfloor \text{ largest of } V_1, \ldots, V_{n+1}.$$

According to our exchangeability assumption over the $n+1$ variables, the right hand side event occurs with at most probability $\lfloor \epsilon \cdot (n+1) \rfloor / (n+1) \leq \epsilon$. □

### A.2 PROOF OF THEOREM 3.3

*Proof.* Once again, this is a well-known result; we prove it here for completeness (see also Tibshirani et al. (2019) for a similar proof). For notational convenience, let random variable $V_i$ be the nonconformity score $\mathcal{S}(X_i, Y_i)$. Since the nonconformity scores $V_i$ are constructed symmetrically, then

$$((X_1, Y_1), \ldots, (X_{n+1}, Y_{n+1})) \stackrel{d}{=} ((X_{\sigma(1)}, Y_{\sigma(1)}), \ldots, (X_{\sigma(n+1)}, Y_{\sigma(n+1)}))$$

$$\Longleftrightarrow (V_1, \ldots, V_{n+1}) \stackrel{d}{=} (V_{\sigma(1)}, \ldots, V_{\sigma(n+1)})$$

for all permutations $(\sigma(1), \ldots \sigma(n+1))$. Therefore, if $\{(X_i, Y_i)\}_{i=1}^{n+1}$ are exchangeable, then so too are their nonconformal scores $\{V_i\}_{i=1}^{n+1}$.

By the construction of $\mathcal{C}_n(X_{n+1})$, we have

$$Y_{n+1} \notin \mathcal{C}_n(X_{n+1}) \Longleftrightarrow \texttt{pvalue}(V_{n+1}, V_{1:n}) \leq \epsilon.$$

Then, using Lemma 3.2, $\mathbb{P}(Y_{n+1} \in \mathcal{C}_n(X_{n+1})) = 1 - \mathbb{P}(Y_{n+1} \notin \mathcal{C}_n(X_{n+1})) \geq 1 - \epsilon$. □

### A.3 PROOF OF THEOREM 4.3

*Proof.* (1) First we prove that $\mathcal{C}_n^{\min}(X_{n+1})$ satisfies Eq. 5. We begin by establishing exchangeability of the minimal nonconformity scores, $\mathcal{S}_g^{\min}(X_i, Y_i)$, as defined in Eq. 6. We simplify notation once again by letting $V_i := \mathcal{S}(X_i, Y_i)$, and $M_i := \mathcal{S}_g^{\min}(X_i, Y_i)$. To denote $\mathcal{S}(X_i, Y)$ for a general random $Y \in \mathcal{Y}$, we write $V_i^{(Y)}$. By the same argument as Theorem 3.3, we have that the base $\{V_i\}_{i=1}^{n+1}$ are exchangeable. Since the label expansion and subsequent $\min$ operator are taken *point-wise* over the $V_i$, we retain symmetry, and therefore

$$(V_1, \ldots, V_{n+1}) \stackrel{d}{=} (V_{\sigma(1)}, \ldots, V_{\sigma(n+1)}) \Longrightarrow (M_1, \ldots, M_{n+1}) \stackrel{d}{=} (M_{\sigma(1)}, \ldots, M_{\sigma(n+1)})$$

for all permutations $(\sigma(1), \ldots \sigma(n+1))$. Thus the $\{M_i\}_{i=1}^{n+1}$ are also exchangeable.

Next, we want to prove that $\mathcal{C}_n^{\min}$ satisfies Eq. 5, $\mathbb{P}\left(|\mathcal{A}_g(X_{n+1}, Y_{n+1}) \cap \mathcal{C}_n^{\min}(X_{n+1})| \geq 1\right) \geq 1 - \epsilon$. Let $|\mathcal{A}_g(X_{n+1}, Y_{n+1})| = k$, where $\mathcal{A}_g(X_{n+1}, Y_{n+1}) = \{\bar{Y}_{n+1}^1, \ldots, \bar{Y}_{n+1}^k\}$. Then:

$$\mathbb{P}\left(|\mathcal{A}_g(X_{n+1}, Y_{n+1}) \cap \mathcal{C}_n^{\min}| \geq 1\right) = \mathbb{P}\left(\bigcup_{i=1}^k \bar{Y}_i \in \mathcal{C}_n^{\min}\right)$$

$$\geq \max\left\{\mathbb{P}\left(\bar{Y}_1 \in \mathcal{C}_n^{\min}\right), \ldots, \mathbb{P}\left(\bar{Y}_k \in \mathcal{C}_n^{\min}\right)\right\}$$

$$= \max\left\{\mathbb{P}\left(\texttt{pvalue}(V_{n+1}^{(\bar{Y}^1)}, M_{1:n}) > \epsilon\right), \ldots, \mathbb{P}\left(\texttt{pvalue}(V_{n+1}^{(\bar{Y}^k)}, M_{1:n}) > \epsilon\right)\right\}$$

$$\stackrel{(i)}{=} \mathbb{P}\left(\texttt{pvalue}(M_{n+1}, M_{1:n}) > \epsilon\right)$$

$$\stackrel{(ii)}{\geq} 1 - \epsilon$$

where $(i)$ is by construction of $M_{n+1}$ (assuming *w.l.o.g.* that it is unique in the case of the random tie-breaking in $\texttt{pvalue}$), and $(ii)$ comes from applying Lemma 3.2 to the exchangeable $\{M_i\}_{i=1}^{n+1}$.

(2) We now prove the second part of Theorem 4.3: that $\mathbb{E}\left[|\mathcal{C}_n^{\min}(X_{n+1})|\right] \leq \mathbb{E}\left[|\mathcal{C}_n(X_{n+1})|\right]$. Again, to simplify notation, we drop the dependence on $(x_i, y_i)$ where obvious.

Let $\mathbf{1}\{y_i \in \mathcal{C}_n\}$ be the indicator random variable that $y_i \in \mathcal{Y}$ is included in $\mathcal{C}_n$. Then:

$$\mathbb{E}\left[|\mathcal{C}_n|\right] = \mathbb{E}\left[\sum_{i=1}^{|\mathcal{Y}|} \mathbf{1}\{y_i \in \mathcal{C}_n\}\right] = \sum_{i=1}^{|\mathcal{Y}|} \mathbb{E}[\mathbf{1}\{y_i \in \mathcal{C}_n\}] = \sum_{i=1}^{|\mathcal{Y}|} \mathbb{P}\left(\texttt{pvalue}(V_{n+1}^{(y_i)}, V_{1:n}) > \epsilon\right).$$

By the same derivation we also have $\mathbb{E}\left[|\mathcal{C}_n^{\min}|\right] = \sum_{i=1}^{|\mathcal{Y}|} \mathbb{P}\left(\texttt{pvalue}(V_{n+1}^{(y_i)}, M_{1:n}) > \epsilon\right).$

Since $M_i \leq V_i \; \forall i \in [1, n+1]$ by definition, we get that $V_{n+1} \leq M_i \Rightarrow V_{n+1} \leq V_i$, and it is the easy to see from the definition of the $\texttt{pvalue}$ (see Eq. 2) that this leads to $\mathbb{P}(\texttt{pvalue}(V_{n+1}^{(y)}, M_{1:n}) > \epsilon) \leq \mathbb{P}(\texttt{pvalue}(V_{n+1}^{(y)}, V_{1:n}) > \epsilon), \forall y \in \mathcal{Y}$. Therefore, it follows that $\mathbb{E}\left[|\mathcal{C}_n^{\min}|\right] \leq \mathbb{E}\left[|\mathcal{C}_n|\right]$. $\quad\square$

### A.4 PROOF OF THEOREM 5.1

*Proof.* We restate our assumption that $\mathcal{M}$ is a valid multiple hypothesis testing correction procedure that properly controls the family-wise error rate,

$$\tilde{P} = \mathcal{M}(P_1, \ldots, P_m) \quad \text{s.t.} \quad \mathbb{P}\left(\tilde{P} \leq \epsilon \mid Y \text{ is correct}\right) \leq \epsilon,$$

and that it is element-wise monotonic[4]

$$(P_1, \ldots, P_m) \preccurlyeq \left(\hat{P}_1, \ldots, \hat{P}_m\right) \implies \mathcal{M}(P_1, \ldots, P_m) \leq \mathcal{M}\left(\hat{P}_1, \ldots, \hat{P}_m\right),$$

where $P_i$ are p-values produced by different nonconformity measures for the same point $(X, Y)$, and $\preccurlyeq$ operates element-wise. First we show that all $\mathcal{C}_n^m$ constructed via Eq. 10 satisfies Eq. 1 when all p-values are known and no pruning has been done. $\mathcal{M}$ operates element-wise over examples, therefore exchangeability is conserved and all basic individual p-value computations from nonconformity scores are valid as before. Then, by the construction of $\mathcal{C}_n^m(X_{n+1})$, we have

$$Y_{n+1} \notin \mathcal{C}_n(X_{n+1}) \iff \tilde{P} \leq \epsilon,$$

and therefore,

$$\begin{aligned} \mathbb{P}\left(Y_{n+1} \in \mathcal{C}_n^m(X_{n+1})\right) &= 1 - \mathbb{P}\left(Y_{n+1} \notin \mathcal{C}_n^m(X_{n+1})\right) \\ &= 1 - \mathbb{P}\left(\tilde{P} \leq \epsilon\right) \geq 1 - \epsilon, \end{aligned}$$

where the final inequality comes from the first assumption on $\mathcal{M}$. Next, we want to prove that early pruning does not remove any candidates $y$ that would *not* be removed from $\mathcal{C}_n^m$. When all p-values after step $j$ are not yet known, we set $P_{k>j}$ to 1. Using the element-wise monotonicity of $\mathcal{M}$, we have $\mathcal{M}(P_1, \ldots, P_j, 1, \ldots 1) \geq \mathcal{M}(P_1, \ldots, P_m)$. Therefore,

$$\left\{y \in \mathcal{Y} \colon \tilde{P}_j^{(y)} > \epsilon\right\} \supseteq \left\{y \in \mathcal{Y} \colon \tilde{P}_m^{(y)} > \epsilon\right\},$$

yielding $y \in \mathcal{C}_n^m(X_{n+1}) \Rightarrow y \in \mathcal{C}_n^j(X_{n+1}), \forall y \in \mathcal{Y}$. We finish by using the earlier result for $\mathcal{C}_n^m(X_{n+1})$ to get $\mathbb{P}\left(Y_{n+1} \in \mathcal{C}_n^j(X_{n+1})\right) \geq \mathbb{P}\left(Y_{n+1} \in \mathcal{C}_n^m(X_{n+1})\right) \geq 1 - \epsilon, \quad \forall j \in [1, m]$. $\quad\square$

## B APPROXIMATE EXPANDED ADMISSION

Assumption 4.1 states that $g$ is a deterministic criterion that always produces a subset of the full ground truth, i.e., $Y_{n+1} \in A_g(X_{n+1}, Y_{n+1}) \subseteq f(X_{n+1})$. As stated earlier, for many tasks, this is a reasonable assumption. For example, $g(X, Y, \bar{Y})$ might be the outcome of sampling a random human annotator, applying a fixed set of valid syntactic normalization rules, or checking if $\bar{Y}$ falls within some $\alpha$-neighborhood of $Y$. Then, by design, $g$ has no error when identifying admissible answers.

In other cases, we may wish to *learn* an approximation $\hat{g}$ of the true admission criterion $g$. For example, for QA, we might be interested in learning a "semantic equivalence" function based on auxiliary paraphrase pairs. This inherently introduces some error, i.e., $\hat{g}(x, y, \bar{y}) \neq g(x, y, \bar{y})$ w.p. $\delta$.

---

[4]As noted in the main text, though we formally require this condition, we are unaware of any common $\mathcal{M}$ beyond contrived examples satisfying Eq. 8 but not Eq. 9. See Appendix E for common MHT corrections.

In the following, we sketch a process for accounting for this error while retaining valid predictions. We leave a more thorough analysis, formal proof, and experimentation for future work.

As the true error rate of $\hat{g}$ is unknown, we can instead bound it on a small set of labelled i.i.d. data,[5] by measuring the empirical false-positve rate $\hat{\gamma}$ over $m$ samples:

$$\hat{\gamma} := \frac{1}{m} \sum_{i=1}^{m} \mathbf{1} \left\{ \underset{\bar{y} \in \mathcal{A}_{\hat{g}}(x_i, \bar{y})}{\arg\min} \ \mathcal{S}(x_i, \bar{y}) \notin \mathcal{A}_{\hat{g}}(x_i, y) \right\}. \tag{11}$$

The empirical error follows a binomial distribution.

Borrowing from the proof of Theorem 4.3, let $\widehat{M}_{n+1}$ be the minimal nonconformity score of the *approximated* admissible answers. We then want to bound $\mathbb{P}(\widehat{M}_{n+1} \leq \tau, \bar{y}^* \in \mathcal{A}_g(X_{n+1}, Y_{n+1}))$, where $\bar{y}^*$ is the most conforming approximate admissible answer used to derive $\widehat{M}_{n+1}$ and $\tau$ is some threshold. If we assume that these joint events are non-negatively dependent (i.e., that our most-conforming labels tend to be ones that the approximate classifier does not make errors on), then we can bound $\mathbb{P}(\widehat{M}_{n+1} \leq \tau, \bar{y}^* \in \mathcal{A}_g(X_{n+1}, Y_{n+1}))$ by $\mathbb{P}(\widehat{M}_{n+1} \leq \tau) \cdot \mathbb{P}(\bar{y}^* \in \mathcal{A}_g(X_{n+1}, Y_{n+1}))$.

The first term can then be bounded in a similar manner as in Lemma 3.2, and the second by binomial confidence intervals for the true error rate $\gamma$, given the observed $\hat{\gamma}$. Then $\tau$ can be set to be the corresponding quantile that allows this probability bound to exceed $1 - \epsilon$.

## C  IMPLEMENTATION DETAILS

**Multiple labeled answers.**    When multiple answers are given (i.e., $y_{n+1}$ is a set) we take $\mathcal{A}_g(x_{n+1}, y_{n+1})$ as the union of all the admissible answers, along with any additional answers expanded by $g$. For the standard conformal prediction baseline, we calibrate on one of the answers at a time, chosen uniformly at random. Note that this is important to preserve equivalent sample sizes across *min*-calibrated CP and standard CP.[6]

**Open-Domain QA.**    We use the open-domain setting of the Natural Questions (NQ) dataset (Kwiatkowski et al., 2019). In this task, the goal is to find a short span from any article in Wikipedia that answers the given question. Questions in the NQ dataset were sourced from real Google search queries, and human annotators identified answer spans to the queries in Wikipedia articles (we use the short answer span setting, and only consider answerable, non-boolean questions).

We use the open-source, pre-trained DPR model for retrieval (i.e., the document retriever) and the BERT model for question answering (i.e., the document reader) provided by Karpukhin et al. (2020). To summarize briefly, the DPR model is trained to maximize the dot-product similarity between the dense representations (obtained via BERT embeddings) of the question and the passage that contains the answer. For candidate passages, we use the Wikipedia-based corpus processed by Karpukhin et al. (2020), where each article is split into disjoint passages of 100 words, resulting in a total of 21,015,324 passages. The DPR model pre-computes dense representations for all passages and indexes them with FAISS (Johnson et al., 2017) for efficient retrieval. At test time, relevant documents for a given dense question encoding are retrieved via fast similarity search. The reader model is a standard BERT model with an independent span prediction head. This model encodes the question and passage jointly, and therefore, the representations cannot be pre-computed. For each token, the model outputs independent scores for being the start or end of the answer span. We also follow Karpukhin et al. (2020) by using the output of the "[CLS]" token to get a passage selection score from the reader model (to augment the score of the retriever). For ease of experimentation, we only consider the top 5000 answer spans per question—taken as the top 100 passages (ranked by the retriever) and the top 50 spans per passage (ranked by the reader). In order to be able to evaluate all $\epsilon \in (0, 1)$ we discard questions whose answers do not fall within this selection. We retain 6750/8757 questions from the validation set and 2895/3610 from the test set.

We compose the cascade for this task using four metrics: the retriever score, followed by the reader's passage selection score, the "span start" score, and the "span end" score. For the non-cascaded models, we take the sum of the span start and span end scores as a single metric.

---

[5]We must slightly strengthen our assumptions from simply exchangeable data to i.i.d. data.

[6]Another possibility would be to calibrate on *all* the given answers, but we found this generally does worse.

| Task | Metric | Description |
|------|--------|-------------|
| **QA** | retriever | $-1\cdot$ similarity score of the question and paragraph by the retrieval model. |
| | passage | $-1\cdot$ logit of the passage selection score by the BERT reader. |
| | span start | $-1\cdot$ logit of the answer's first token by the BERT reader. |
| | span end | $-1\cdot$ logit of the answer's last token by the BERT reader. |
| | span sum | span start + span end. |
| **IR** | BM25 | $-1\cdot$ BM25 similarity score between query and candidate. |
| | CLS logit | $-1\cdot$ logit of the claim-evidence pair by the ALBERT classifier. |
| **DR** | RF | $-1\cdot$ score of the candidate by the Random Forest model. |
| | MPNN | $-1\cdot$ score of the candidate by the directed Message Passing Neural Network. |

Table C.1: Nonconformity measures used in our experiments.

**IR.** We use the FEVER dataset for evidence retrieval and fact verification (Thorne et al., 2018). We focus on the retrieval part of this task. Note that the retrieved evidence can then be used to verify the correctness of the claim automatically (Nie et al., 2019; Schuster et al., 2019), or manually by a user. We follow the dataset splits of the Eraser benchmark (DeYoung et al., 2020) that contain 97,957 claims for training, 6,122 claims for validation, and 6,111 claims for test. The evidence needs to be retrieved from a set of 40,133 unique sentences collected from 4,099 total Wikipedia articles.

We compose the cascade using two metrics: an efficient non-neural BM25 model, and a neural sentence-pair classifier. Our BM25 retriever uses the default configuration available in the Gensim library (Řehůřek and Sojka, 2010). We perform simple preprocessing to the text, including removing punctuation and adding word stems. We also add the article title to each sentence. Our neural classifier (CLS) is built on top of ALBERT-Base and is trained with BCE on (claim, evidence) pairs. We collect 10 negative pairs for each positive one by randomly selecting other sentences from the same article as the correct evidence. For shorter articles, we extend the negative sampling to also include the top (spurious) candidates indentified by the BM25 retriever.

**DR.** We construct a molecular property screening task using the ChEMBL dataset (see Mayr et al., 2018). Given a specified constraint such as "*is active for property A and property B but not property C*", we want to retrieve at least one molecule from a given set of candidates that satisfies this constraint. The motivation of this task is to simulate in-silico screening for drug discovery, where it is often the case where chemists will searching for a new molecule that satisfies several constraints (such as toxicity and efficacy limits), out of a pool of many possible molecular candidates.

We split the ChEMBL dataset into a 60-20-20 split of molecules, where 60% of molecules are separated into a train set, 20% into a validation set, and 20% into a test set. Next, we take all properties that have more than 1000 labeled molecules (of which at least 100 are positive and 100 are negative, to avoid highly imbalanced properties). Of these $\sim$200 properties, we take all N choose K combinations that have at least 100 molecules with all K properties labelled (ChEMBL has many missing values). We set K to 3. For each combination, we randomly sample an assignment for each property (i.e., $\{\text{active, inactive}\}^K$). We keep 5000 combinations for each of validation and test sets. The molecules for each of the combinations are only sourced from their respective splits (i.e., molecular candidates for constraints in the property combination validation split only come from the molecule validation split). Therefore, at inference time, given a combination we have never seen before, on a molecules we have never seen before, we must try to retrieve at least one molecule that has the desired combination assignment.

Both our random forest (RF) and the directed Message Passing Neural Network (MPNN) were implemented using the `chemprop` repository (Yang et al., 2019). The RF model is based on the Scikit library (Pedregosa et al., 2011) and uses 2048-bit binary Morgan fingerprints (Rogers and Hahn, 2010) of radius 2 to independently predict all target properties (active or inactive) for a given molecule. The RF model is fast to run during inference, even on a single CPU. The MPNN model uses graph convolutions to learn a deep molecular representation, that is shared across property predictions. Each property value (active/inactive) is predicted using an independent classifier head. The final prediction is based on an ensemble of 5 models, trained with different random seeds. Given a combination assignment $(Z_1 = z_1, \ldots, Z_k = z_k)$, for both the RF and MPNN models, we take

the nonconformity score as the model's negative log-likelihood, where the likelihood is computed independently, i.e. $p_\theta(Z_1 = z_1, \ldots, Z_k = z_k) = \prod p_\theta(Z_i = z_i)$.

# D   ADDITIONAL EXPERIMENTAL RESULTS

We provide supplemental experimental results to those shown in §6. In D.1 we show an example of a *closed-domain* QA task where cascading multiple measures actually *boosts* the predictive efficiency to the extent that it outweighs the conservative MHT correction factor. In D.2 we compare our method to heuristic methods at fixed efficiency levels, as described in §7.

## D.1   COMPLEMENTARY CONFORMAL CASCADES FOR CLOSED-DOMAIN QA

The primary motivation in this work for conformalized cascades is to improve *computational efficiency* by allowing cheaper models to pre-filter the candidate space prior to running more expensive and more powerful models. Though not guaranteed, in some cases it is also possible that combining different nonconformity scores together has an overall synergistic effect that outweighs the generally conservative effects of the MHT corrections. This is similar in theory to ensembles or mixtures-of-experts (Jacobs et al., 1991), and similar results have been reported for combined conformal prediction (Carlsson et al., 2014; Linusson et al., 2017; Toccaceli and Gammerman, 2017).

While the tasks in §7 focus on cascades that are designed primarily for efficiency, here we also explore cascades for the smaller-scale task of closed-domain questions answering on the SQUAD 2.0 dataset (Rajpurkar et al., 2018). This task isolates the *document reader* aspect of the open-domain QA pipeline, in which a relevant passage is already given. We cascade two primary models: (1) a span extractor (EXT) that gives independent scores for the start and end positions of the answer span, and (2) a more expensive answer classifier (CLS) that considers the entire candidate span (i.e., models the start and end jointly). We briefly outline the implementation details before giving the results below.

The EXT model uses the ALBERT-Base QA model (Lan et al., 2020). It is trained to maximize the likelihood of the answer span $[i, j]$, where $p(\text{start} = i, \text{end} = j)$ is modeled as $p(\text{start} = i)p(\text{end} = j)$. During inference, the model computes all $\mathcal{O}(n^2)$ start and end position scores, and predicts the pair with the highest sum. We use the start and end position scores as two separate nonconformity measures. The CLS model is also built on top of ALBERT-Base, and is similar to Lee et al. (2016). Instead of scoring start and end positions independently, we concatenate the corresponding hidden representations at tokens $i$ (start) and $j$ (end), and score them jointly using an MLP. We then train with binary cross-entropy (BCE) over correct and incorrect answers. We limit the number of negative samples (incorrect answers) to the top 64 incorrect predictions of the EXT model.

The authors keep the original test set hidden; for ease of CP-specific evaluation we re-split the development set randomly by article. This results in 5,302 questions in our validation set, and 6,571 questions in our test set. The average length of a paragraph in the evaluation set is 127 words. Together with the "no answer" option, a question for a paragraph of that length will have 8,129 candidate answer spans. For the purposes of experimentation, we filter out questions for which the EXT model does not rank the correct answer within the top 150 predictions (so we can compute the full $(0, 1)$ range of $\epsilon$ tractably). This discards less than 0.5% of questions.

Our results across several values of epsilon are given in Table D.1. As in the tasks in §7, the *min*-calibrated CPs greatly improve over the baseline CP. In this case, however, the cascaded CP also consistently outperforms the CP with only a single measure.

| $1 - \epsilon$ | CP | | *min* CP | | Cascaded *min* CP | | |
|---|---|---|---|---|---|---|---|
| | Succ. | $|\mathcal{C}_n|$ | Succ. | $|\mathcal{C}_n^{\min}|$ | Succ. | $|\mathcal{C}_n^{\min}|$ | Amortized cost |
| 0.99 | 1.00 | 17.69 | 0.99 | 6.68 | 0.99 | 6.31 | 0.50 |
| 0.95 | 0.98 | 5.14 | 0.95 | 3.14 | 0.96 | 2.45 | 0.42 |
| 0.90 | 0.95 | 3.09 | 0.90 | 1.98 | 0.92 | 1.76 | 0.40 |
| 0.80 | 0.86 | 1.66 | 0.80 | 1.31 | 0.83 | 1.24 | 0.38 |

Table D.1: CP results on the SQUAD 2.0 dataset.

## D.2 HEURISTIC METHODS

In addition to evaluating our improvements over regular conformal prediction, we compare our conformal method to other common heuristics for making set-valued predictions. Specifically, we consider baseline methods that given some scoring function $\texttt{score}(x, y)$ and threshold $\tau$ to define the output set of predictions $\mathcal{B}$ at $x \in \mathcal{X}$ as

$$\mathcal{B}(x, \tau) := \{y \in \mathcal{Y} \colon \texttt{score}(x, y) \geq \tau\}, \tag{12}$$

where $\tau$ is then tuned on the calibration set to find the largest threshold for the desired accuracy:

$$\tau_\epsilon^* := \sup \left\{ \tau \colon \frac{1}{n} \sum_{i=1}^{n} \mathbf{1}\{y_i \in \mathcal{B}(x, \tau)\} \geq 1 - \epsilon \right\}. \tag{13}$$

The prediction for the test point $x_{n+1}$ is then $\mathcal{B}(x_{n+1}, \tau_\epsilon^*)$. We consider two variants: (1) fixed top-$k$, where $\texttt{score} := -\mathrm{rank}(\texttt{metric}(x_{n+1}, y))$ according to some metric, and (2) raw thresholding, where $\texttt{score} := \texttt{metric}(x_{n+1}, y)$, i.e., some raw, unnormalized metric. Top-$k$ is simple and relatively robust to the variance of the metric used, but as it doesn't depend on $x$, it also means that both easy examples and hard examples are treated the same (giving prediction sets that are too large in the former, and too small in the latter). Raw metric thresholding, on the other hand, gives dynamically-sized prediction sets, but is more sensitive to metric variance. We emphasize that these baselines do not provide theoretical coverage guarantees for potentially non-i.i.d. finite samples.

When ignoring smoothing factors and restricting ourselves to a single, non-cascaded metric, it is straightforward to see that choosing the nonconformity measure to be $\mathcal{S} := \mathrm{rank}(\texttt{metric}(x, y))$ or simply $\mathcal{S} := -\texttt{metric}(x, y)$ makes the set $\mathcal{C}_n(x_{n+1})$ equivalent to $\mathcal{B}(x_{n+1}, \tilde{\tau}_\epsilon)$, where

$$\tilde{\tau}_\epsilon := \mathrm{Quantile}\left(1 - \epsilon, \widehat{F}\left(\{\mathcal{S}(x_1, y_1), \dots, \mathcal{S}(x_n, y_n)\} \cup \{\infty\}\right)\right). \tag{14}$$

We write $\widehat{F}(\mathcal{V})$ to denote the empirical distribution of set $\mathcal{V}$. For large enough $n$, $\tilde{\tau}_\epsilon$ becomes nearly identical to $\tau_\epsilon^*$. Note, however, that this comparison only applies to the *split* (i.e., inductive) conformal prediction setting. Metrics computed without relying on a held-out calibration set (as *full* CP is able to do) must treat the $n + 1$ data points symmetrically, a key feature of CP. Our methods extend to multiple cascaded metrics, finite $n$, and the full CP setting.

We compare predictive efficiency results when using top-$k$ and threshold heuristics versus our CP methods in Figure D.1 and Table D.2. As expected, with large enough $n$, the baseline CP and Threshold methods are nearly equivalent. In some cases, top-$k$ outperforms the threshold- and CP-based methods that use raw scores (this is likely due to variance in the densities of high scoring candidates across examples). When applying optimizing for admissible answers, our *min*-calibrated CP improves over all three methods. Note that optimizing for admissible answers is also applicable for the heuristic methods, in which case the trends will be similar to that of the baseline CP.

# E   MULTIPLE HYPOTHESIS TESTING

As we discuss in §5, naively combining multiple hypothesis tests will lead to an increased family-wise error rate (FWER). For a visual example, see the uncorrected cascaded CP (blue dashed lined) in Figure E.1. It demonstrates that combining nonconformal measures without using a MHT correction can result in accuracies smaller than $1 - \epsilon$ (i.e., invalid coverage).

Several methods exist for correcting for MHT by bounding the FWER (with different assumptions on the dependencies between the hypothesis tests). We experiment with the Bonferroni and Simes procedures. We test the corrections on each task's validation set and find Simes to work well for QA and IR, but not to hold for DR (See Figure E.1). Therefore, we use the Bonferroni correction for DR and Simes for the other two tasks. For completeness, we briefly describe the two methods and the assumptions they rely on below (extensive additional details can be found in the literature).

## E.1   BONFERRONI PROCEDURE

The Bonferroni correction is easy to apply and does not require any assumptions on the dependencies between the hypothesis tests used (e.g., independent, positively dependent, *etc*). However, it is generally very conservative. The Bonferroni correction scales $\epsilon$ by a factor of $1/m$, and uses the

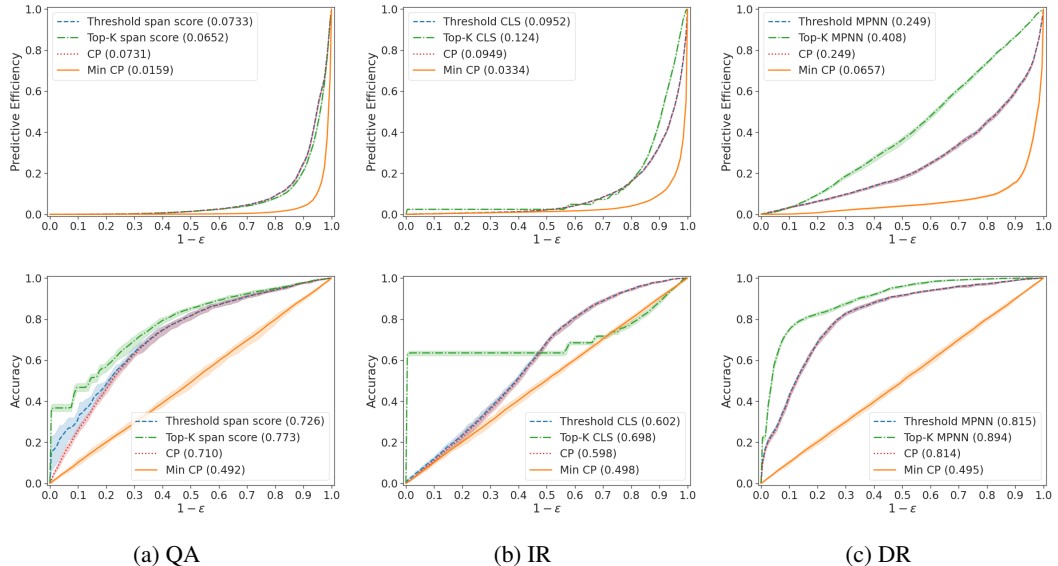

(a) QA (b) IR (c) DR

Figure D.1: *How does our conformal method compare to non-conformal baselines?* We show predictive efficiency and accuracy across various target accuracies (values of $1 - \epsilon$). As expected, with a large enough calibration set, the performance of the threshold heuristic is similar to CP. The top-$k$ threshold is harder to tune, resulting in lower than the desired accuracy for some values of IR, and sometimes in unnecessary large prediction sets. Specific values of $\epsilon$ are compared in Table D.2.

| Task | Target Acc. | Threshold | | Top-$k$ | | Baseline CP | | *min*-CP | |
|------|-------------|-----------|---------|---------|---------|-------------|-------------|----------|----------------------|
| | | Acc. | $\|\mathcal{B}\|$ | Acc. | $\|\mathcal{B}\|$ | Acc. | $\|\mathcal{C}_n\|$ | Acc. | $\|\mathcal{C}_n^{\min}\|$ |
| **QA** | 0.90 | 0.98 | 1243.8 | 0.98 | 1041.7 | 0.98 | 1245.7 | 0.90 | 198.0 |
| | 0.80 | 0.94 | 452.8 | 0.95 | 390.5 | 0.94 | 453.5 | 0.80 | 58.7 |
| | 0.70 | 0.91 | 228.8 | 0.92 | 203.3 | 0.91 | 227.8 | 0.70 | 22.1 |
| | 0.60 | 0.87 | 128.4 | 0.89 | 120.5 | 0.87 | 127.8 | 0.60 | 10.6 |
| **IR** | 0.99 | 1.00 | 33.6 | 1.00 | 41.0 | 1.00 | 33.6 | 0.99 | 17.8 |
| | 0.95 | 1.00 | 20.9 | 0.95 | 29.8 | 1.00 | 20.9 | 0.95 | 7.4 |
| | 0.90 | 0.98 | 13.8 | 0.89 | 18.7 | 0.98 | 13.8 | 0.90 | 4.0 |
| | 0.80 | 0.95 | 6.7 | 0.78 | 6.7 | 0.95 | 6.7 | 0.80 | 1.7 |
| **DR** | 0.90 | 0.99 | 429.5 | 1.00 | 652.9 | 0.99 | 429.8 | 0.90 | 84.1 |
| | 0.80 | 0.97 | 305.8 | 1.00 | 525.1 | 0.97 | 305.8 | 0.80 | 42.7 |
| | 0.70 | 0.96 | 216.8 | 0.99 | 398.3 | 0.96 | 216.6 | 0.70 | 28.1 |
| | 0.60 | 0.94 | 145.7 | 0.98 | 266.5 | 0.94 | 145.6 | 0.60 | 19.3 |

Table D.2: Non-CP baseline results on the test set for different target accuracy values. We compare the predictive efficiencies of the CP methods ($\|\mathcal{C}_n\|$ and $\|\mathcal{C}_n^{\min}\|$) to those of the baseline methods ($\|\mathcal{B}\|$).

union bound on the p-values $(P_1, \ldots, P_m)$ to get:

$$\mathbb{P}\left(Y_{n+1} \notin \mathcal{C}_n^m(X_{n+1})\right) = \mathbb{P}\left(\bigcup_{i=1}^{m} P_i \leq \frac{\epsilon}{m}\right) \leq \sum_{i=1}^{m} \mathbb{P}\left(P_i \leq \frac{\epsilon}{m}\right) = \epsilon.$$

We achieve the same result by scaling each p-value by $m$, and take the final combined p-value to be the minimum of all the scaled p-values. It is easy to show that this correction is monotonic.

### E.2 Simes Procedure

The Simes procedure (Simes, 1986) allows a stricter bound on the FWER when the measures are multivariate totally positive (Sarkar and Chang, 1997), which usually holds in practice (Rødland, 2006). To apply the correction, we first sort the p-values in ascending order, and then perform an order-dependent correction where the correction factor decreases as the index of the p-value increases.

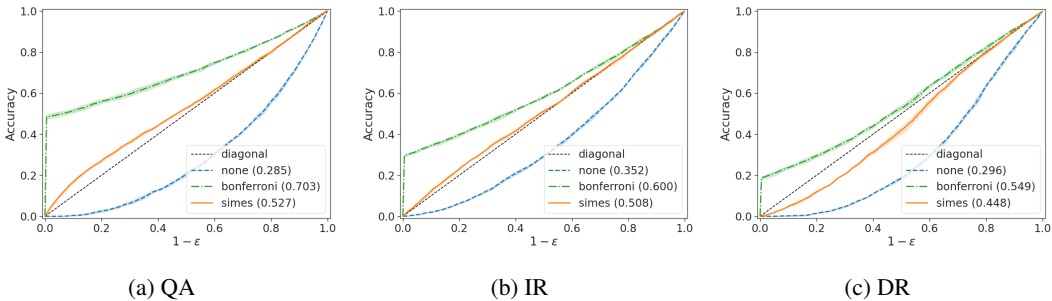

(a) QA           (b) IR           (c) DR

Figure E.1: Success rate against tolerance thresholds with different methods for MHT correction (validation set). Not applying any correction leads to a in-valid classifier (the success rate is below the diagonal). The Bonferroni method is conservative, leading to a valid classifier, but sometimes with a higher accuracy rate than necessary (a result of having excessively large $\mathcal{C}_n$). The Simes correction works for the QA and IR tasks and provides a tighter bound for them. The Simes correction does not work for the DR task, likely due to a violation of its MTP2 assumption, but the Bonferroni method provides a relatively tight correction there—especially for small $\epsilon$.

Specifically, if $(P_{(1)}, \ldots P_{(m)})$ are the sorted p-values $(P_1, \ldots P_m)$ in an $m$-level cascade, we modify the p-values to be $P_{(i)}^{\text{Sim}} = m \cdot \frac{p_{(i)}}{i}$. We take the final combined p-value to be the minimum of the corrected p-values. It is easy to show that this correction is monotonic.

