# OpenReview forum: "Efficient Conformal Prediction via Cascaded Inference with Expanded Admission"
_ICLR.cc/2021/Conference — ICLR 2021 Poster_

### Official Review · AnonReviewer2 · 2020-10-28
**The paper offers an interesting extension of conformal prediction set framework and provide valuable computational methodologies. Some imprecisions need to be clarified but the overall contributions is well presented and interesting for the ICLR conference.**

**Rating:** 6
**Confidence:** 3

**Review:**

The paper propose a conformal prediction (CP) approach in which the prediction output is not necessarily unique; and will be assumed to belong to a finite set. Then, the authors analyze how the classical CP set can be modified wrt to a relaxed requirement that it is sufficient to contains an admissible answer. This lead to a smaller set in expectation and a coverage guarantee is established, proving the validity of the method. Authors also provide a computational methodology to screen out non promising candidate. The paper is well written and provide practical numerical experiments that demonstrate usefulness of the introduced CP in "real" world.

- Assumption 4.1 should be probabilistic since $A_g(X, Y)$ is a random set. Is the inclusion required *almost surely*? Note also that ${\bar Y^j}$ is not defined here. It is also unclear how the sequence of admissible answers $(\bar Y_{i}^{j})_{j \in [k], i \in [n]}$ is included in the datasets (both training and testing). This leads to the following confusion

- In Eq. 1, P is the distribution of $(X, Y)$. However in Eq. (5), it seems that P is the distribution of $(X, {\bar Y^j}_{j \in [k]})$ (whose sample is assumed exchangeable).

- Computational complexity of CP:
The comment in the beginning of section 5 seems inexact: the cost of computing CP is not linear in $|\mathcal{Y}|$ otherwise it would be impossible to compute in regression case where $\mathcal{Y}$ is the real line $\mathbb{R}$.

- How this cascade strategy could be used in regression setting? For instance, despite the limitation of statistical estimator that can be used, full conformal prediction set can be computed by using parametric programming (without loss in the coverage guarantee):
(J. Lei 2019): Fast exact conformalization of Lasso using piecewise linear homotopy.
(Ndiaye and Takeuchi, 2019): Computing Full Conformal Prediction Set with Approximate Homotopy. For instance in the latter, one could consider cascade parameterized by the optimization error. How this cascade approach can improve the computational efficiency of full CP in regression?

- In section 2, CP paragraph, these two papers might be relevant:
(Chernozhukov etal 2019): Distributional conformal prediction. And (Kivaranovic etal 2019): Adaptive, distribution-free prediction intervals for deep neural networks

- In section 3, the notion of conformity measure is not well defined (S must be symmetric wrt permutation of the data).

- epsilon in (0, 1) in the introduction.

- Given g, *x and y*, any prediction that is a member ...

- Is Eq. 5 equivalent to $P(\exists j \in [k]: \bar Y_{n+1}^{j} \in C(X_{n+1})) \geq 1 - \epsilon$? (which seems more explicit)

---

> ### Author Response · Authors · 2020-11-14
> **Review response**
>
> We thank the reviewer for the thoughtful review and helpful comments. We respond to individual comments below. The manuscript will be updated shortly to reflect any indicated changes (we will notify when this happens).
>
> **On the distribution $P$:**
> We respond to this question first, as it will help clarify the subsequent points.
>
> As we describe in Section 4, we assume an underlying set-valued function $f: \mathcal{X} \rightarrow 2^\mathcal{Y}$ that expresses the full set of admissible answers for input $X$. This function remains unknown and what we observe in our dataset is rather a single sample, $(X_i, Y_i)$, from this underlying ground truth set via some additional observation process (e.g., influenced by which annotator wrote the answer to the question).
>
> Therefore $P_{X, Y}$ governing each pair $(X_i,Y_i)$ is an induced distribution from this set-valued function together with the unknown observation process. $\bar{Y}_i^j \, j \in [k]$ notation was intended to refer to the underlying set, i.e., $\bar{Y}_i^j \, j \in [k] = f(X_i)$. We agree that the notation is not very clear and we’ll update the text to use the underlying set valued function $f(X_i)$ directly.
>
> **On Assumption 4.1:**
> $A_g(X_i, Y_i)$ is deterministic for any fixed $(X_i,Y_i)$. The randomness is therefore the same as in $(X_i,Y_i)\sim P$, and arises from the ground truth answers directly.
>
> To clarify further, as we define it in this work, $A_g$ is the result of a deterministic expansion via the user-defined admission function $g$---and is therefore deterministic w.r.t. the random sample $(X_i, Y_i)$. The caveat here is that $A_g$ defines the “reachable” subset of the ground truth $f(X_i)$ given a sample of an admissible label $Y_i$. For example, applying syntactic normalization rules to a sample reference for a question from SQuAD will yield more, but still not necessarily *all*, of the possible admissible answers.
>
> **On Eq. 5:**
> Strictly speaking, no. Eq. 5 is taken w.r.t. $A_g$, which is a stricter statement. We aren’t measuring whether *any* admissible answer is included in $C(X_{n+1})$, rather we are requiring it to be one that we can also verify using $g$ and the sample reference $Y_i$.
>
> Of course, the probability expressed (in the reviewer’s notation) is greater than the one in Eq. 5, and is therefore also $\geq 1 - \epsilon$.
>
> **On conformal regression:**
> In this work we focus only on conformal classification, for problems with large (but finite) label sets. Regression is indeed another interesting and important setting that we will be happy to address in future work.
>
> **Other:**
> Thank you for the various suggestions and corrections. We’ll provide a formal definition of the conformity measure S (including its symmetry condition), and add the suggested references to the related work.

---

> > ### Author Response · Authors · 2020-11-22
> > **Addressed comments**
> >
> > Dear R2,
> >
> > Our new draft now contains what we think is a much clearer Section 4 (addressing P and assumption 4.1, etc). We have also updated the related work to include the suggested references.
> >
> > Please let us know if we have addressed your concerns, or if there are any others that you might have.

---

### Official Review · AnonReviewer1 · 2020-10-28
**Solid experimental results, moderate significance/novelty, some presentation improvements suggested**

**Rating:** 6
**Confidence:** 3

**Review:**

Summary:

This paper presents two advances in conformal prediction, a field with information retrieval applications in which a set of candidate responses to a query is presented and the objective is to return a small set of responses with at least one of the responses being the correct response.  The first contribution is a method in which the possibility of several admissible responses is modeled (rather than there being just one response) with the system being calibrated against the odds that a particular response is the "most admissible", i.e. most conforming to the joint query/response distribution being learned from data. The second contribution is a cascaded prediction system in which simpler and less computationally expensive models are used for initial filtering and then more sophisticated and more expensive models are used downstream to further refine the response set. Rigorous statistical adjustments are used to account for multiple-hypothesis-test issues arising from using the cascading system.

Pros:

Reasonably thorough experimental results demonstrating performance gains in terms of sensitivity/specificity as well as in terms of computational cost are presented.

The paper is mostly well written and the theory is presented with a good amount of rigor.

Cons:

I feel that the cascading technique is only of moderate significance. It's a good idea and there's some value in promoting it, but it also feels like a solution that might get arrived at by a savvy, practical-minded machine learning engineer in industry rather than a particularly profound ML research concept. The rigorous multiple hypothesis corrections (Bonferroni , etc) are certainly appreciated and are something which a less sophisticated practitioner might not know to apply, so that aspect of it feels more like an academic-paper-level contribution, but the basic idea of cascading feels like an applied, industry systems solution that someone with common sense might arrive at on her/his own.

Further suggestions:

I strongly suggest changing the definition of "predictive efficiency" on page 6.  Defining it such that lower "efficiency" is better will confuse many readers. Efficiency has a well-established common-language meaning as something which is desirable. Why not define the efficiency as (for instance) the percentage of the entire candidate Y set which is eliminated, so that higher efficiency is better?

I also think that it would broaden the appeal and accessibility of the paper to spend more time relating conformal prediction to more widely known information retrieval concepts like precision and recall and maybe also learning-to-rank. I suspect far more readers will be familiar with basic information retrieval, precision, recall, etc than with conformal prediction. I acknowledge that one of the appendices connects conformal prediction back to some of these concepts. I think maybe some of that appendix content belongs in the main paper.

---

> ### Author Response · Authors · 2020-11-14
> **Review response**
>
> We thank the reviewer for the helpful comments and suggestions. We respond to individual comments below. The manuscript will be updated shortly to reflect any indicated changes (we will notify when this happens).
>
> **Contribution of the conformalized cascade:**
> While the idea of cascades is not new, cascades are both effective and address a real practical need in our setting when the number of possible outcomes is large. Our specific use cases are also a bit different from prior art and we provide rigorous guarantees for our cascade.
>
> **Predictive efficiency terminology:**
> We agree with the reviewer that higher efficiency would be a more natural reading. In this paper, we chose to follow the standard definition used in the conformal prediction literature, as put forth by the seminal work of Vovk. et. al., 2005 (Algorithmic Learning in a Random World). We also note that this definition is easy to quickly translate to the absolute average number of retrieved candidates (i.e., efficiency * |Y|), which is also a helpful, practical metric. We understand, however, that this metric can be counter-intuitive, and will try to clarify this early on so as to remove any confusion.
>
> **Relation to information retrieval metrics:**
> We thank the reviewer for the suggestion. In the case of standard conformal prediction, where Y is a *single* answer, indeed it is possible to similarly express our objective in IR terms. In this case, we would be interested in maximizing *precision* while holding the level of *recall* above $1 - \epsilon$.
>
> However, when moving to the case of conformal prediction with expanded admission (i.e., where there is a set of admissible answers, but we only require one), we would have to slightly modify the standard definitions of “recall” and “precision”. We will mention the connection to precision and recall with respect to standard CP when discussing the IR task in our Introduction.
>
> With respect to ranking, at least for the purposes of this work, we treat the prediction set $C(X_{n+1})$ as “unordered”. That is, we do not consider any ranking (the ranking is implicitly expressed via the nesting property created by increasing/decreasing epsilon).

---

> > ### Author Response · Authors · 2020-11-22
> > **Addressed comments**
> >
> > Dear R1,
> >
> > In our new draft we included more connections to precision and recall as explanation in our introduction. We also hope that our comments here have addressed concerns about the significance of our contribution w.r.t. the cascade.
> >
> > Please let us know if we have addressed your concerns, or if there are any others that you might have.

---

> > > ### Comment · AnonReviewer1 · 2020-11-24
> > > **Thanks for the changes/clarifications**
> > >
> > > Thanks for adding the explanation of the connection to precision/recall as well as the explanation that the definition of efficiency is already established in this subfield. Much appreciated.

---

> > > > ### Author Response · Authors · 2020-11-24
> > > > **Thank you for the review**
> > > >
> > > > We are glad that the explanation has helped make the paper clearer! Do let us know if there are any further concerns we can try to address.

---

### Official Review · AnonReviewer5 · 2020-11-08
**Interesting and exciting direction, proposed method comes with theoretical guarantees and boasts empirical efficacy on large scale applications**

**Rating:** 8
**Confidence:** 3

**Review:**

Conformal prediction (CP) allows for the selection of a set of candidate answers guaranteed to contain the correct answer with some probability. The authors propose two extensions to CP, 1. To extend validity for all admissible answers, 2. Using prediction cascades to improve computational efficiency. The authors show that their approaches provide similar guarantees on accuracy like CP but with lowered predictive efficiency and computational cost.


Reasons to accept:
1.  The approaches are simple, novel, and interesting and they come with theoretical guarantees
2. The proposed methods allow for CP to be extended to some realistic tasks
3. Impressive results showing lowered predictive efficiency and computational cost
4. The paper was well written and the approaches and experiments seem technically sound

Overall, I believe that this would be a useful paper for the community, and based on the reasons given above I would recommend acceptance.

---

> ### Author Response · Authors · 2020-11-14
> **Review response**
>
> We thank the reviewer for the helpful comments.

---

### Author Response · Authors · 2020-11-22
**Updated Revision**

Dear Reviewers,

Thank you for your helpful comments. We have now uploaded a revision that we hope addresses the raised concerns, particularly about clarity and presentation.

---

### Decision · Program_Chairs · 2021-01-07
**Final Decision**

**Decision:**

Accept (Poster)

**Comment:**

This paper presents an approach for conformal prediction where, in its standard paradigm, a set of prediction candidates is identified as opposed to a single one.  The authors advance the CP framework by presenting a rigorous methods that allows for a smaller set of admissable predictions with a covergae quarantee. Their further contribution is a methodolgy based on cascading that filters out non promising candidates.

After the discussion period, _all_ the reviewers are in favour of accepting the manuscript with the average being marginally above acceptance.  My recommendation is therefore to accept the paper.

Strong points:
The advance of a smaller set of admissible predictions in the CP framewrok is quite useful especially in scenaria where the set can grow (expensively) large.
Thorough experimental analysis with good presentation of performance gain and usefulness in real world data.

Weak points:
Lack of novelty in the techniques make the work a weaker candidate compared to the rest of submissions.